

# Retrieval of aerosol properties from ceilometer and photometer measurements: long-term evaluation with in-situ data and statistical analysis at Montsec (southern Pyrenees)

Gloria Titos[1*], Marina Ealo[1,2], Roberto Román[3,4], Alberto Cazorla[3], Yolanda Sola[2], Oleg Dubovik[5], Andrés Alastuey[1], Marco Pandolfi[1]

[1]Institute of Environmental Assessment and Water Research (IDAEA), CSIC, Barcelona, Spain
[2]Group of Meteorology, Department of Applied Physics, Faculty of Physics, University of Barcelona, Spain
[3]Andalusian Institute for Earth System Research, IISTA-CEAMA, Granada, Spain
[4]Grupo de Óptica Atmosférica (GOA), Universidad de Valladolid, Spain
[5]Laboratoire d'Optique Atmosphérique (LOA). Université Lille, France
[*]Now at: Applied Physics Department, University of Granada, Spain

*Correspondence to*: Gloria Titos (gtitos@ugr.es)

**Abstract.** Given the need of accurate knowledge of aerosol microphysical and optical properties with height resolution, various algorithms combining vertically-resolved and column integrated aerosol information have been developed in the last years. Here we present new results of vertically-resolved extensive aerosol optical properties (backscattering, scattering and extinction) and volume concentrations retrieved with the GRASP (Generalized Retrieval of Aerosol and Surface Properties) algorithm over a 3 year-period. The range-corrected signal (RCS) at 1064 nm measured with a ceilometer and the aerosol optical depth (AOD) and sky radiances from a sun/sky photometer have been used as input for this algorithm. We perform a detailed evaluation of GRASP retrievals with simultaneous in-situ measurements performed at the same height, at the Montsec mountaintop observatory (MSA) in the Pre-Pyrenees (northeastern Spain). This is the first long-term evaluation of various outputs of this algorithm; previous evaluations focused only on the study of aerosol volume concentration for short-term periods. In general, our results show good agreement between techniques although GRASP inversions yield higher values than those measured in-situ. The statistical analysis of the extinction coefficient vertical profiles shows a clear seasonality as well as significant differences depending on the air-masses origin. The observed seasonal cycle is mainly modulated by a higher development of the atmospheric boundary layer (ABL) during warm months, which favors the transport of pollutants to MSA, and higher influence of regional and North-African episodes. On the other hand, in winter, MSA is frequently influenced by free troposphere conditions and venting periods, and therefore lower extinction coefficients that markedly decrease with height. This study shows the potentiality of implementing GRASP in ceilometers and lidars networks for obtaining aerosol optical properties and volume concentrations at multiple sites worldwide, which will definitely contribute to enhance the representativeness of aerosol vertical distribution as well as to provide useful information for satellite and global models evaluation.





## 1 Introduction

Atmospheric aerosol particles interact directly with the incident solar radiation by either scattering or absorbing light. These aerosol-radiation as well as the aerosol-cloud interactions influence the Earth's radiative budget and therefore have an impact on climate. Aerosol particles are considered the atmospheric constituents with the largest uncertainty in global climate forcing estimations (IPCC, 2013). Their high spatial, vertical and temporal variability is one of the key factors contributing to their large uncertainty (IPCC, 2013).

During the last years, a great effort has been done from the Aerosols, Clouds, and Trace gases Research InfraStructure (ACTRIS, www.actris.eu) community to extend the temporal and spatial coverage of aerosol properties sampling and to harmonize measurement protocols to increase their representativeness and the comparability among sites and between measurement techniques (i.e. in-situ versus remote-sensing). In-situ observatories are widely distributed and cover a large variety of atmospheric conditions (urban, rural, background and remote sites). Moreover, in-situ instrumentation is able to provide a complete set of information in terms of chemical, optical and microphysical aerosol properties. The main drawback of in-situ observatories is that they are only representative of the atmospheric layer closest to the surface and might not be useful to infer aerosol radiative properties at elevated layers (Rosati et al., 2016). For this reason, vertically resolved aerosol observations are needed to complement surface in-situ measurements and column-integrated observations from satellites or ground-based photometers. Lidar systems are frequently used for profiling aerosol optical properties and, depending on the lidar capabilities and availability of co-located photometer measurements, vertical profiles of aerosol microphysical properties can be retrieved as well by using inversion algorithms (Chaikovsky et al., 2008, 2016; Lopatin et al., 2013).

One of the recently developed inversion algorithms is the Generalized Retrieval of Aerosol and Surface Properties (GRASP; Dubovik et al., 2014; www.grasp-open.com) code that uses the heritage of AERONET (Aerosol Robotic Network) inversion scheme (e.g. Dubovik and King, 2000; Dubovik et al., 2006). It is a versatile and open-source algorithm capable of obtaining optical and microphysical aerosol properties from different sets of measurements (Kokhanovsky et al., 2015; Espinosa et al., 2017; Torres et al., 2017; Román et al., 2017; 2018). In particular, GRASP allows the user to combine Aerosol Optical Depths (AODs), sky radiances and range corrected signal (RCS) lidar values to retrieve columnar and vertically-resolved aerosol properties. Román et al. (2018) proposed a similar approach but using the RCS values at only one wavelength measured with ceilometer instead of using multi-wavelength lidar RCS values as done before. The retrieved vertical profiles of aerosol volume concentration showed good agreement with in-situ measurements from an aircraft campaign and with in-situ measurements from a nearby mountain station during a summer campaign in southern Spain (Román et al., 2018). The use of ceilometer measurements in the GRASP algorithm is a significant advance towards a better representation of aerosol properties with vertical resolution since ceilometers are cheaper, require less supervision and are more extensively distributed compared to more sophisticated lidar systems (Cazorla et al., 2017; Dionisi et al., 2018). GRASP profiles from ceilometers worldwide could be used for evaluating dust forecast models (Tsekeri et al., 2017) such as the BSC-DREAM8b, as input to radiative transfer models (Granados-Muñoz et al., 2019) or can be assimilated in global models (Chen et al., 2018). This application represents



a step-forward in the classical use of ceilometers that were originally developed for cloud base detection (e.g., Martucci et al., 2010).

The potential of this new technique motivates the present study in which the GRASP code is used to retrieve long-term vertical profiles of aerosol optical and microphysical properties combining ceilometer and AERONET sun/sky photometer measurements over a 3-year period. The main objective of this paper is to evaluate the performance of the retrieved aerosol products by GRASP combining ceilometer and photometer measurements using as reference the in-situ measurements performed at the Montsec Global Atmosphere Watch (GAW) station (MSA, 1570 m a.s.l., NE Spain). Additionally, a statistical analysis of the vertical structure of aerosol properties based on the 3-years of GRASP retrievals at MSA is presented.

## 2 Experimental site and instrumentation

### 2.1 Montsec Observatory

Measurements used in this study were performed in the northeastern Iberian Peninsula, most of them at the Montsec mountain-top station (MSA; 42° 3' N, 0° 44' E, 1570 m a.s.l.), located in the facilities of the Montsec Astronomic Observatory (OAdM, http://oadm.ieec.cat/). The MSA continental background site is part of the Catalan Air Quality Network (Xarxa de Vigilància i Previsió de la Contaminació Atmosfèrica, http://territori.gencat.cat/) and it is integrated in the European Research Infrastructure ACTRIS and in the Global Atmosphere Watch (GAW) program. It is a remote high-altitude station situated in the southern side of the Pre-Pyrenees at the Montsec d'Ares mountain. This region is sparsely populated and isolated from large urban and industrial agglomerations (140 km from Barcelona to the northwest and 30 km from the largest city in the region). The prevailing atmospheric conditions are characteristic of Mediterranean climate, with long dry periods, sporadic but intense rains, and a prevalence of local and regional atmospheric air mass circulations and Saharan dust intrusions (Ripoll et al., 2014; Ealo et al., 2016).

### 2.2 In-situ measurements

Aerosol particles light scattering ($\sigma_{sp}$) and hemispheric backscattering ($\sigma_{bsp}$) coefficients were measured at three wavelengths (450, 525 and 635 nm) with a LED-based integrating nephelometer (model Aurora 3000, ECOTECH Pty, Ltd, Knoxfield, Australia) with 5-min time resolution. The aerosol flow in the nephelometer was set to 5 lpm. Measurements were performed at dry conditions (RH<40%) by using the internal RH-control function of the nephelometer that slightly heats the sampled air when the RH is above the threshold value. The nephelometer is periodically calibrated (four times per year) with $CO_2$ and filtered air. Zero adjustments are performed every midnight using internally filtered particle free air. The Aurora 3000 nephelometer used in this study operates by collecting light scattered within the angular range 10–171° (Müller et al., 2011a). The main source of error is the truncation in the forward direction (0–10°) due to the inability of the nephelometer to sense near-forward scattering, which is an increasingly dominant part of the total scattering for large particles (Anderson et al., 1996). Non-idealities due to truncation errors have been corrected following the scheme described by Müller et al. (2011a). The



detection limits of the nephelometer over 1 min averaging time are 0.11, 0.14 and 0.12 Mm$^{-1}$ for total scattering at 450, 525 and 635 nm, respectively, and 0.12, 0.11 and 0.13 Mm$^{-1}$ for backscattering (Müller et al., 2011a).

The aerosol light-absorption coefficient, $\sigma_{ap}$, was measured with a Multi-Angle Absorption Photometer (MAAP, model 5012, Thermo) at 637 nm (Müller et al., 2011b). A detailed description of the method is provided by Petzold and Schönlinner (2004).

The MAAP draws the ambient air at constant flow rate of 16.7 lpm and provides 1 min values. The detection limit of the MAAP instrument is lower than 0.6 Mm$^{-1}$ over 2 min integration. The total method uncertainty for the particle light-absorption coefficient inferred from MAAP measurements is around 12% (Petzold and Schönlinner, 2004).

An aerosol optical counter (GRIMM Spectrometer, model 1129-Sky-OPC) was used to measure particle number concentrations in 31 size bins, for particles in the diameter size range from 0.25 to 32 µm at 5 min time resolution. The working

principle of this instrument is based on multi-channel light-scattering optics (Grimm and Eatough, 2009) in which the intensity of the measured scattered light is related to the size of the particles. Volume size distributions were derived from the number size distribution assuming spherical particles.

All in-situ measurements were performed at MSA station and have been referred to ambient temperature and pressure using the measurements from an automatic and collocated weather station. Measurements were performed at low relative humidity

(RH<40%), as recommended by the World Meteorological Organization (WMO/GAW, 2003) and ACTRIS infrastructure.

### 2.3 Passive remote sensing measurements

Measurements of column integrated aerosol properties were determined with a CE-318 sun/sky photometer (Cimel Electronique, France) included in AERONET (Holben et al., 1998) and located at MSA observatory. This instrument performs direct sun measurements with a 1.2º full field of view at least at 440, 675, 870, and 1020 nm, which are used to derive AOD

at these wavelengths. The sky radiance measurements (almucantar configuration) are also carried out at 440, 675, 870 and 1020 nm. A full description of the AERONET products obtained from this instrument can be found in Holben et al. (1998). In this work, AOD and sky radiances, both at 440, 675, 870 and 1020 nm, from version 2 of AERONET level 1.5 data are used.

### 2.4 Active remote sensing measurements

Vertical profiles of RCS at 1064 nm were performed with a Jenoptik CHM 15k Nimbus (G. Lufft Mess- und Regeltechnik

GmbH, Germany) ceilometer that includes a pulsed Nd:YAG laser, emitting at 1064 nm. The energy emitted per pulse is 8 µJ and the duration of each pulse is between 1 and 5 ns with a repetition frequency of 6.5 kHz. The maximum height of the signal is 15.36 km a.g.l. equivalent to 1024 range bins. The ceilometer is located 760 meters downslope of the MSA measurement station, at the Center for the Observation of the Universe (COU, http://www.parcastronomic.cat/). The horizontal distance between the ceilometer and the MSA station is less than 2.5 km. This instrument operates continuously with a temporal

resolution of 1 min and a spatial resolution of 15 m. According to the manufacturer overlap function, the overlap of the telescope and the laser beam is greater than 85% at around 760 m a.g.l. Nevertheless, the RCS profiles provided by the instrument are overlap-corrected using the manufacturer's overlap function.



## 3 GRASP retrievals

GRASP code is mainly based on two independent modules: 1) the forward module consisting of a radiative transfer and aerosol model which simulates the radiative measurements for a given aerosol scenario, and 2) the numerical inversion module which is not related to the physical nature of the inverted data (Dubovik et al., 2011; Dubovik et al., 2014) and is mathematically
based on multi-term least square method (Dubovik and King, 2000). Detailed description of the GRASP working principle using sun/sky photometer and RCS data can be found in Lopatin et al. (2013), where the GARRLiC (Generalized Aerosol Retrieval from Radiometer and Lidar Combined data) scheme, which is part of GRASP code, is explained.

In this study, we follow the inversion strategy named as $GRASP_{pac}$ (sub-index meaning "photometer and ceilometer") introduced by Román et al. (2018). A $GRASP_{pac}$ retrieval is done for each sky radiance almucantar sequence available from
AERONET if sky radiances and ceilometer measurements satisfy cloud-free conditions. The following measurements are used in GRASP code for each retrieval: 1) the cloud-screened sky radiance and AOD at 440, 675, 870 and 1020 nm (AERONET version 2 level 1.5); and 2) the normalized ceilometer RCS at 1064 nm, previously cloud-screened, smoothed and averaged in a ±15 min window centered in the photometer measurement time, at 60 log-spaced heights as in Lopatin et al. (2013). The minimum height of these 60 values corresponds to the MSA altitude while the maximum height could be up to 7000 m above
MSA. In addition, Bidirectional Reflectance Distribution Function (BRDF) is needed to make the $GRASP_{pac}$ retrievals and, to this end, an 8-days climatology (2000-2014) of the MCD43C1 product (V005 MODIS Terra+Aqua BRDF/Albedo 16-Day L3 0.05Deg CMG) of MODIS (MODerate-resolution Imaging Spectroradiometer) is used (Schaaf et al., 2011).

Since ceilometer measurements are limited to a single wavelength, it is not possible to vertically differentiate between aerosol modes/types and therefore vertical profiles of intensive variables such as the single scattering albedo (SSA), lidar ratio (LR)
or effective radius are assumed as vertically constant by this method. As a result, for each $GRASP_{pac}$ retrieval we obtain aerosol profiles (at 60 points) of backscattering, scattering, extinction and absorption coefficients at 440, 675, 870, 1020 and 1064 nm, and also of aerosol volume concentration and size distribution.

## 4 Results and discussion

### 4.1 $GRASP_{pac}$ – in-situ comparison

#### 4.1.1 Optical properties comparison

In-situ extensive aerosol optical properties (i.e. backscattering, scattering and extinction coefficients) measured at MSA over a 3 years period (April 2014 - March 2017) were used for evaluating the retrieval of aerosol optical properties from a ceilometer and a sun/sky photometer using the $GRASP_{pac}$ method in a long-term frame. Previous evaluations of this algorithm with in-situ data focused on aircraft campaigns (2-3 study cases) (e.g. Benavent-Oltra et al., 2018; Tsekeri et al., 2017) or short-term
periods (Román et al., 2018). Figure 1 shows the comparison between the $GRASP_{pac}$ retrievals and in-situ measured coefficients at low ambient RH ($RH_{ambient} < 50\%$). This restriction has been imposed to avoid cases affected by hygroscopic





growth and consequent enhancement of the optical coefficients detected by the remote sensing instrumentation. To merge both datasets (GRASP$_{pac}$ and in-situ), the data have been averaged in 1-hour intervals. Figure 2 shows the relative differences between optical parameters measured in-situ and retrieved by GRASP$_{pac}$ optical parameters. In general, the GRASP$_{pac}$ retrievals are in agreement with the in-situ measurements. The coefficients of determination span from 0.49 for the

backscattering coefficient to 0.77 for the scattering coefficient and 0.73 for the extinction coefficient (see details in Figure 1). For the backscattering coefficient comparison, it is important to bear in mind the intrinsic differences among the variables compared. The nephelometer design limits the collection of the scattered light to the angular range 10-171º, while the ceilometer measures the backscatter signal at 180º. Additionally, for the backscattering coefficient at 180º retrieved with GRASP$_{pac}$ we assume for the comparison with in-situ data that the hemispheric backscattered radiation is symmetric along all

hemispheric angles, which might not be true for all cases and might contribute to lower the correlation coefficient. For both the aerosol light scattering and the extinction coefficients the slope and intercept of the regression are > 1, while for the backscattering coefficient the slope is < 1. As it can be seen in Fig. 2, GRASP$_{pac}$ tends to overestimate all the studied coefficients with higher occurrence of negative relative errors. In this sense, all frequency distributions are tailed towards negative values. The agreement of GRASP$_{pac}$ with in-situ data shows relative differences of ±1.25% for 84%, 75% and 68%

of the backscattering, scattering and extinction coefficients, respectively. For the extinction coefficient, Herreras et al. (2018) showed good agreement between the integrated extinction profiles derived with GRASP$_{pac}$ and AOD from sunphotometers located at various heights (R$^2$>0.6).

[Figure 1]

[Figure 2]

Figure 3 shows the relationship between the scattering and extinction coefficients measured in-situ and retrieved by GRASP$_{pac}$. The color scale represents the difference in the single scattering albedo measured in-situ and retrieved with GRASP$_{pac}$. For the in-situ data, there is a linear trend between scattering and extinction coefficients (R$^2$ = 1), denoting that the aerosol light-extinction is dominated by the scattering process, which is in accordance with previous in-situ studies performed at MSA (Pandolfi et al., 2014). On the contrary, for the GRASP$_{pac}$ retrievals the correlation is also good but the data points deviate

from the 1:1 line as the difference in the SSA between in-situ and GRASP$_{pac}$ increases (yellowish colors). In general, GRASP$_{pac}$ retrievals yield lower SSA values (average SSA of 0.88 ± 0.14) compared with in-situ SSA (0.93 ± 0.04). These discrepancies in the absorption could be related to the differences in the SSA at ground level (as measured in-situ) and the SSA associated with the total atmospheric column (GRASP$_{pac}$) due to absorbing aloft layers. However, the largest disagreement (yellowish colors in Fig. 3b) coincide with Atlantic air masses influence, that as it will be shown in Section 4.2, are characterized by low

aerosol load and low impact of decoupled aerosol layers. On the other hand, Andrews et al. (2017) showed a systematic difference in the SSA from AERONET retrievals compared with integrated in-situ profiles, revealing that AERONET retrievals yield higher aerosol absorption than in-situ measurements, especially at low aerosol load. MSA is a remote site with predominantly low aerosol load and low contribution of absorbing particles. Furthermore, Román et al. (2018) found with synthetic data that SSA values retrieved by GRASP$_{pac}$ reproduce better the real SSA values for moderate-high aerosol loads.





In a similar way, AERONET, in version 2, only provides SSA values with quality assurance if the AOD at 440 nm is higher than 0.4 (Dubovik et al., 2000; Dubovik et al., 2002; Holben et al., 2006). Then, most of the obtained SSA differences could be associated with the low aerosol load conditions, where the SSA uncertainty is high in GRASP$_{pac}$ values.

[Figure 3]

## 4.1.2 Volume size distribution comparison

Figure 4 shows the comparison of the total aerosol volume concentration (V) determined with GRASP$_{pac}$ and measured in-situ at MSA height over the study period. The color scale represents the ratio V$_{fine}$/V that quantifies the contribution of fine particles (diameter below 1 µm) to the total volume concentration. As we can see in Fig. 4a, there is a lack of correlation, showing different relationship depending on the ratio V$_{fine}$/V. When fine particles predominate (i.e. V$_{fine}$/V > 0.75, yellowish colors) the volume concentration measured in-situ is significantly larger than the volume concentration retrieved from the ceilometer and photometer data using GRASP$_{pac}$. On the contrary, when coarse particles predominate the volume concentration provided by GRASP$_{pac}$ is larger than the one determined in-situ. Limiting the comparison to those cases with V$_{fine}$/V < 0.75 (Fig. 4b) improves significantly the correlation ($R^2 = 0.65$), and shows relative differences within ±2% for 90% of the data (Fig. 4c). Similar to the extinction and scattering coefficients comparison, GRASP$_{pac}$ retrievals yield higher aerosol volume concentrations compared with the in-situ measurements. Similar overestimations comparing GRASP$_{pac}$ and in-situ data have been reported before). In particular, Román et al. (2018) compared the GRASP$_{pac}$ retrievals using also ceilometer and photometer data as input with in-situ measurements performed in a mountain station located ~25 km apart from the ceilometer and at around 2000 m above it during an intensive field campaign. Their results show that GRASP$_{pac}$ overestimates the volume concentration with a slope of the comparison around 1.5. We found similar results, revealing that in general, GRASP$_{pac}$ overestimates the aerosol volume concentration (slope of the comparison of 1.34). However, the comparison between GRASP$_{pac}$ and in-situ measurements shows significant discrepancies when fine particles predominate (V$_{fine}$/V > 0.75). The reduced number of cases with V$_{fine}$/V > 0.75 (~15% of the total number of data points) makes it difficult to draw conclusive results concerning the total volume concentration in atmospheric conditions dominated by fine particles. Previous evaluations of GRASP algorithm were mainly conducted during Saharan dust events with predominance of coarse mode particles. Benavent-Oltra et al. (2018) found similar coarse volume concentrations between GRASP retrievals and in-situ profiles during two flights performed under dust-dominated conditions, with slight underestimation of GRASP in the aloft dust plumes, while significant overestimation was reported for the fine volume concentration. Overestimation of fine volume concentrations obtained with GARRLiC (Generalized Aerosol Retrieval from Radiometer and Lidar Combined data algorithm) algorithm compared with in-situ data were also observed under a dust-dominated and a marine polluted cases (Tsekeri et al., 2017). Using synthetic data, Román et al. (2018) showed higher discrepancies in the retrieval of fine volume concentrations than in coarse ones for GRASP$_{pac}$. The reason behind these differences was partly attributed to the use a long wavelength (1064 nm) as RCS in the retrieval which is less sensitive to fine particles than shorter wavelengths. Nevertheless, despite the differences among studies, all of them evidence that the retrieval of fine volume concentrations is particularly challenging while good



results can be obtained for the coarse volume concentration or total concentration if the size distribution is dominated by coarse particles.

[Figure 4]

Finally, several environmental and topographic factors can be brought forward to partly explain the differences observed
among techniques, namely the measurement atmospheric conditions (temperature, pressure and RH) and orographic effects affecting wind patterns and atmospheric boundary layer structure and causing spatial inhomogeneities. Concerning the atmospheric conditions at which the aerosol properties are measured in terms of temperature, pressure and relative humidity, we expect low effect on the comparison since the in-situ data has been converted to ambient T and P and the comparison was restricted to cases with ambient RH < 50%. Although hygroscopic growth can occur even at low RH (Zieger et al., 2017), we
limit the study to ambient RH < 50% in order to minimize the RH effect in the comparison (Titos et al., 2016). As can be seen in Figure S1 of the supplementary material, the comparison shows no dependency on RH for $RH_{ambient} < 50\%$. Another possible factor that could affect the comparison is the fact that the in-situ and photometer measurements are not performed exactly over the ceilometer vertical. However, due to the short horizontal distance (< 2.5 km), this fact is expected to have little impact in our results.

**4.2 Statistical analysis of aerosol profiles**

In the following section, we focus on the extinction coefficient since it is the most relevant climate variable from the ones retrieved with $GRASP_{pac}$. Qualitatively speaking, the volume concentration and the scattering coefficient show similar trends in the vertical distribution. Figure 5 shows the seasonality of particle extinction profiles retrieved with $GRASP_{pac}$ using ceilometer and photometer data as inputs. It is important to recall that $GRASP_{pac}$ retrievals are performed only during daytime
and clear sky conditions (see Section 3 for further details), which might bias the statistical analysis presented in this section compared to continuous measurements. Figure S2 of the supplementary material shows the frequency distribution of the number of profiles retrieved by month and hour of the day. As it can be seen, the $GRASP_{pac}$ retrievals are restricted to daytime conditions and solar zenith angles above 40º (mainly from 6 to 9 h in the morning and from 14 to 16 h). Accordingly, there are also less $GRASP_{pac}$ retrievals during autumn and winter.

In average terms, the largest extinction coefficients are observed at the lowest altitudes sounded. A nearly exponential decrease with height of the median extinction coefficients is observed during all seasons up to 4000-5000 m a.s.l. Exponential decreasing trend of the extinction coefficient has been observed in several statistical lidar studies in Europe (Mattis et al., 2004; Amiridis et al., 2005; Navas-Guzmán et al., 2013). There is a clear seasonal behavior in the vertical distribution of aerosol particles, evidencing that during winter most particles are confined to the first few kilometers above the surface while the median profile
in summer shows the presence of particles at higher altitudes. Also in summer, the extinction profiles display a larger interquartile range compared with the other seasons denoting high variability in the vertical distribution of aerosol particles. Concerning the extinction coefficients in the lowermost part of the profiles, Pandolfi et al. (2014) reported a similar seasonality





for continuous in-situ measurements at MSA, with the highest extinction coefficients observed in summer and the lowest ones in winter.

[Figure 5]

Air masses arriving at MSA have been classified in four sectors following the procedure of Ripoll et al. (2014): Atlantic (ATL),

North-African (NAF), Regional (REG) and European and Mediterranean (EU+MED). Figure 6 shows a statistical overview of the extinction profiles from GRASP$_{pac}$ classified according with their air mass origin. There are significant differences in the extinction vertical distribution depending on the origin of the air masses affecting the Montsec area. The lowest median extinction coefficient occurs under Atlantic air masses. This result is in agreement with the low extinction coefficients found in winter, given that during colder months the site is frequently affected by Atlantic air masses and is located within the free

troposphere (Ripoll et al., 2014). These profiles also show low variability (smaller interquartile range). A similar behavior is obtained for the MED+EU sector, although the extinction coefficient displays higher variability; especially pronounced close to the surface (high 90th percentile). For air masses with origin in North Africa the extinction coefficient vertical profiles show the highest variability; denoting the strong variation in intensity and aerosol-layers stratification among events. The average extinction coefficient for the lowest atmospheric layer is slightly lower than the average extinction coefficient found during

dust events at surface level in MSA using in-situ techniques (Pandolfi et al., 2014). This discrepancy can be attributed to the different study period and therefore different NAF episodes included in the calculation with varying intensity and frequency. The air-masses grouped in the REG sector include transport from the Iberian Peninsula as well as re-circulation processes associated with the land-sea breezes regime (Millán et al., 1997). In this case, the extinction coefficient profiles show high variability up to 6000 m a.s.l., indicating layering and accumulation of pollutants under regional re-circulation conditions.

During these episodes, pollutants are raised up to upper levels resulting in the stratification of aerosol layers along the vertical atmosphere (Pérez et al., 2004). On the other hand, MED+EU and ATL sectors show a low 90th percentile and interquartile range above 3000 m a.s.l., suggesting that the likelihood of aloft aerosol layers under these atmospheric scenarios is significantly reduced compared with REG sector and, more remarkable, with NAF sector.

The air mass classification and the seasonality of the extinction vertical profiles are clearly linked. NAF and REG episodes are

more frequent during spring and summer while ATL episodes are more frequent in autumn and winter (Ripoll et al., 2014). The seasonal cycle observed is mainly modulated by a higher development of the ABL during warm months, and higher influence of REG and NAF episodes (e.g. Ealo et al., 2018). This combination leads to high extinction coefficients at higher altitudes and strong variability (large difference in the 10th and 90th percentiles and interquartile range) during warmer months. However, in winter, MSA is frequently influenced by free troposphere conditions and venting periods (Ripoll et al., 2014),

and therefore lower extinction coefficients. NAF episodes also affect MSA during winter (i.e. Titos et al., 2017), but their frequency of occurrence is low and their impact in the extinction vertical profile is not observed in the median and 90th percentile profiles (Fig. 6).

[Figure 6]





Figure 7 shows the center of mass calculated for the median extinction profile, and the 25th and 75th percentile extinction profiles following the procedure described by Cazorla et al. (2017), as a function of the air mass origin sector. The highest center of mass is achieved under NAF air masses, evidencing the influence of aloft dust layers. During an intense dust outbreak in February 2016, Cazorla et al. (2017) calculated a center of mass of 3000 m a.s.l. at MSA in the most intense day. An

interesting feature of Fig. 7 is the difference in the centers of mass retrieved from the percentiles and median profiles for REG and NAF sectors, while for ATL and MED+EU the difference in the 25th and 75th percentiles is small. This fact evidences the high variability in vertical distribution of aerosol particles occurring during NAF and REG episodes.

[Figure 7]

## 5 Conclusions

In this study, we present a systematic application of the GRASP algorithm using ceilometer RCS and sun/sky photometer measurements (GRASP$_{pac}$) over an extended period of time (3 years). Our unique experimental set-up allows us to perform a long-term evaluation of the GRASP$_{pac}$ retrievals versus in-situ measurements under different atmospheric conditions. The output variables studied here are the aerosol backscattering, scattering and extinction coefficients and the volume concentration. The results show an overall good agreement between GRASP retrievals and in-situ measurements, especially

good for scattering and extinction coefficients ($R^2 > 0.7$). The volume concentration comparison shows differences depending on the predominance of fine or coarse particles, with poor agreement when the contribution of fine particles to the total volume concentration is $> 75\%$, and good agreement otherwise. Restricting the comparison to cases with $V_{fine}/V < 0.75$, GRASP$_{pac}$ and in-situ measurements show good correlation although GRASP$_{pac}$ yield higher volume concentrations. Similar overestimation of GRASP$_{pac}$ is found for the scattering and extinction coefficients. We found slight discrepancies in the

scattering-extinction relationship obtained with GRASP$_{pac}$ compared to in-situ data. In general, GRASP$_{pac}$ retrievals yield lower SSA values (average SSA of $0.88 \pm 0.14$) compared with in-situ SSA ($0.93 \pm 0.04$). This result can be linked with previous evaluations of AERONET retrievals that were shown to yield higher aerosol absorption than in-situ measurements, especially at low aerosol load. Evaluation of GRASP$_{pac}$ algorithm at different environments with variable aerosol load and SSA characteristics will contribute to better understand and constrain the validity and limitations of GRASP$_{pac}$.

The statistical analysis of the extinction coefficient vertical profiles retrieved with GRASP$_{pac}$ shows a clear seasonality as well as significant differences depending on the air-masses origin. The observed seasonal cycle is characterized by higher extinction coefficients during summer with strong day to day variability while during winter the extinction coefficient is lower in the whole atmospheric column and shows lower variability. Similar seasonal behavior is obtained from the in-situ measurements at ground level. This seasonality is associated with a higher development of the atmospheric boundary layer during warm

months, favoring the transport of pollutants to MSA. Additionally, the higher influence of regional and North-African episodes in summer contributes to the observed seasonality. On the other hand, in winter, MSA is frequently influenced by free troposphere conditions and venting periods, and therefore lower extinction coefficients that markedly decrease with height.



The use of automated lidars and ceilometers systems for the determination of vertically-resolved aerosol properties has increased in recent years thanks to their low operation requirements and costs, and their capability of providing continuous unattended measurements. Together with this increase use of ceilometer systems, there is a growing need of being able to convert the ceilometer signals into usable aerosol properties. In this context, the overall good results obtained in our validation

are encouraging and emphasize the potentiality of implementing GRASP in ceilometers and lidars networks for obtaining aerosol optical properties and volume concentrations with height resolution and wide spatial coverage. Compared with previous studies, the  present evaluation of GRASP$_{pac}$ retrievals with in-situ data has been performed over a 3 year-period, being therefore representative of varying atmospheric conditions. The implementation of GRASP$_{pac}$ in the frame of measurement networks will enhance the representativeness of aerosol vertical distribution providing useful information for

satellite and models evaluation, contributing to the objectives of several international initiatives such us the EU COST Action TOPROF (Towards operational ground-based profiling with ceilometers, Doppler lidars and microwave  radiometers  for improving  weather  forecasts) or the E-PROFILE program of the European Meteorological  Services  Network.

**Authors contribution**

GT analyzed the data and wrote the manuscript, ME operated the MSA in-situ station, RR performed the GRASP retrievals,

AC processed the ceilometer data in the frame of ICENET, YS operated the sun/sky photometer at MSA, OD provided feedback on the GRASP algorithm, AA designed the experiment, MP operated the ceilometer, designed the experiment. All authors provided comments on the manuscript.

**Acknowledgments**

This work was supported by the MINECO (Spanish Ministry of Economy and Competitiveness, and FEDER funds, under the

CTM2015-66742-R and PRISMA and HOUSE projects (CGL2012-39623-C02/00, CGL2016-78594-R), the MAGRAMA (Spanish Ministry of Agriculture, Food and Environment), the Generalitat de Catalunya (AGAUR 2017 SGR0041 and the Departament de Territori I Sostenibilitat). This work has received funding from the European Union's Horizon 2020 research and innovation programme under grant agreement No 654109 (ACTRIS-2). M. Pandolfi is funded by a Ramón y Cajal Fellowship (RYC-2013-14036) awarded by the MINECO. R. Román is funded by MINECO under post-doctoral programme

Juan de la Cierva – Incorporación (IJCI-2016-30007). G. Titos is funded by MINECO under post-doctoral programme Juan de la Cierva (FJCI-2014-20819 and IJCI-2016-29838). We thank the OAdM and COU astronomical observatories for their support. Thanks are due to AERONET and RIMA networks for the scientific and technical support. The MODIS MCD43C1 data product was retrieved from the online Data Pool, courtesy of the NASA Land Processes Distributed Active Archive Center (LP DAAC), USGS/Earth Resources Observation and Science (EROS) Center. The authors acknowledge the use of GRASP





inversion algorithm (www.grasp-open.com), and also thank D. Fuertes, A. Lopatin and B. Torres for their feedback in the use of GRASP.

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





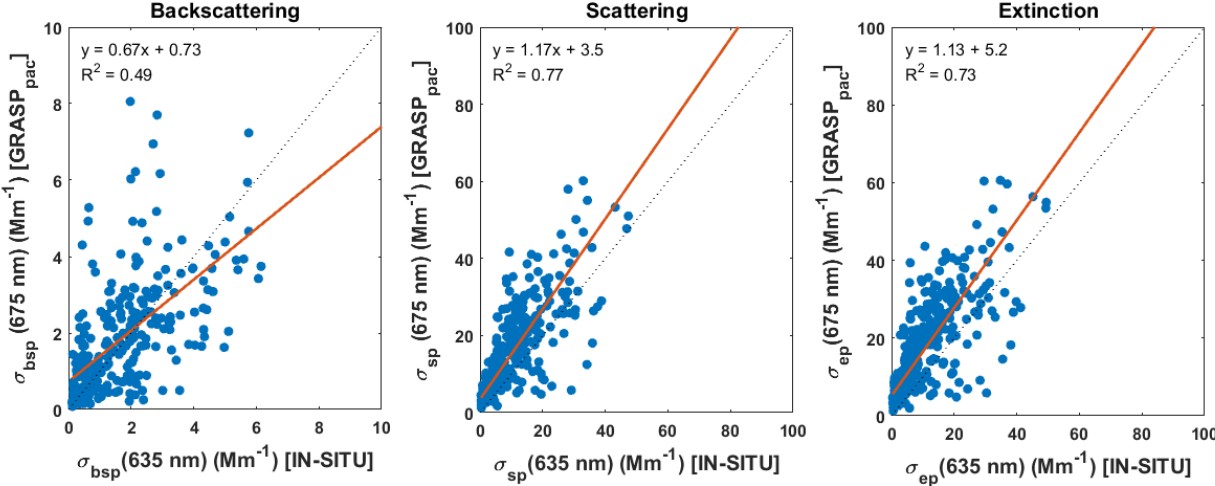

**Figure 1: Scatter plots of the hourly averaged aerosol light backscattering, scattering and extinction coefficients determined with GRASP_pac from the ceilometer and photometer data at MSA height versus the measured in-situ coefficients. This comparison is restricted to situations with low ambient RH (RH_ambient < 50%). The linear regression and the 1:1 line are also shown.**





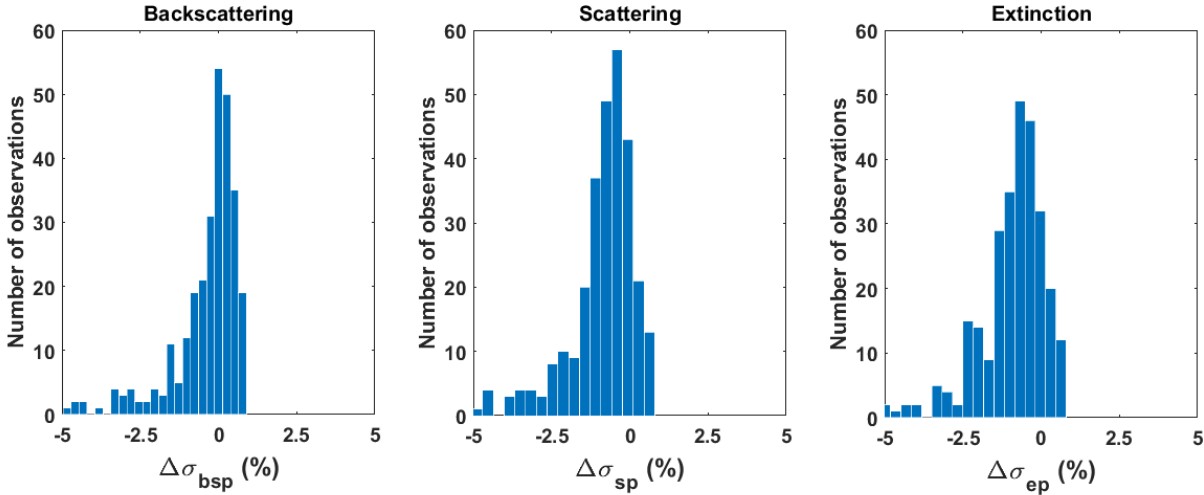

**Figure 2: Histograms of the relative difference between in-situ measured and retrieved with GRASP_pac optical parameters (aerosol light backscattering, scattering and extinction coefficients) at low ambient RH (RH_ambient < 50%).**





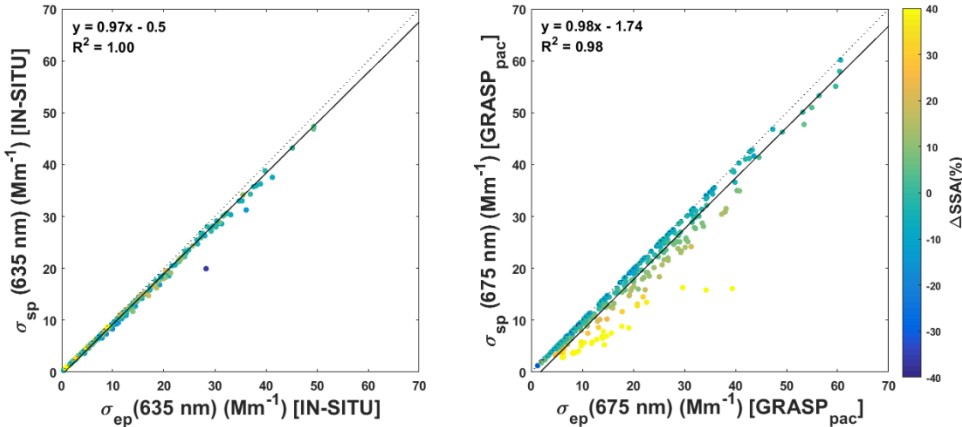

**Figure 3: Scatter plots of the hourly averaged aerosol light scattering and extinction coefficients measured in-situ (left panel) and retrieved with GRASP$_{pac}$ algorithm (right panel). The color scale represents the relative difference in the single scattering albedo, SSA, between in-situ and GRASP$_{pac}$ data.**




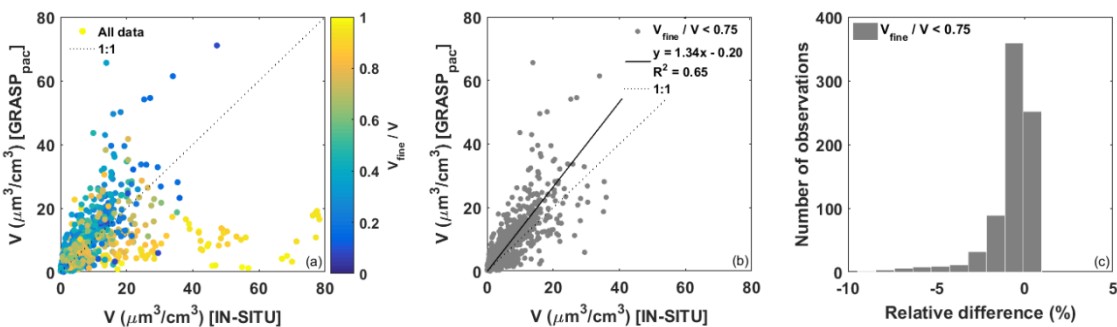

**Figure 4: (a) Scatter plot of the hourly averaged aerosol volume concentration determined with GRASP_pac from the ceilometer and photometer data at MSA height versus the in-situ concentrations at low ambient RH (RH_ambient < 50%), with the color scale representing the contribution of fine particles to the total volume concentration. (b) Same than (a) but restricted to situations with contribution of fine particles to the total aerosol volume concentration < 75% (V_fine / V < 0.75). (c) Frequency of occurrence of the relative difference between the volume concentrations measured in-situ and determined with GRASP_pac for situations with V_fine / V < 0.75.**





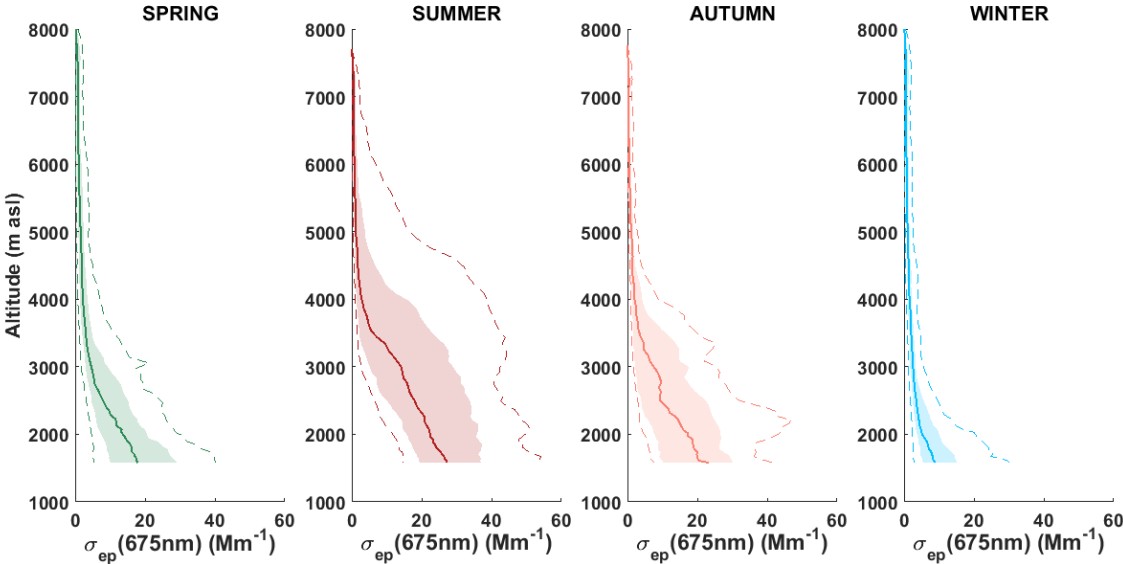

**Figure 5: Seasonal variability of vertical profiles of aerosol light-extinction coefficients at 675 nm. The line represents the median and the shadowed area is the interquartile range. The dashed-lines represent the 10th and 90th percentiles. Seasonal statistics are based on daily averaged profiles. Spring corresponds with March, April and May; Summer with June, July and August; Autumn with September, October and November; Winter with December, January and February.**



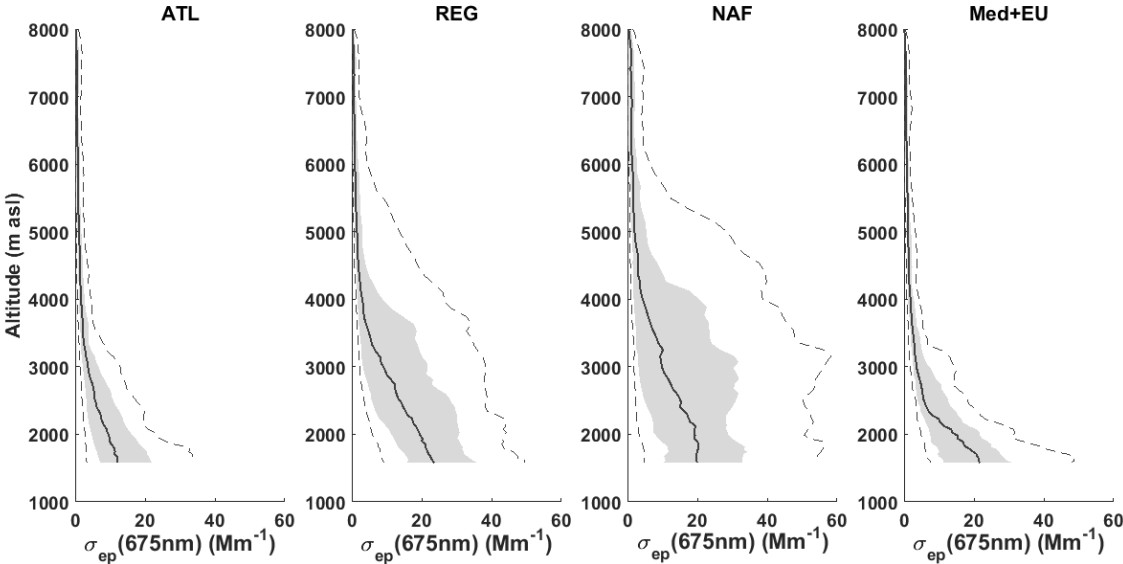

**Figure 6: Particle light-extinction coefficient profiles at 675 nm classified by air mass origin (ATL = Atlantic, REG = Regional, MED+EU = Mediterranean and European, NAF = North African). The line represents the median and the shadowed area is the interquartile range. The dashed-lines represent the 10th and 90th percentiles. Statistics are based on daily average profiles.**



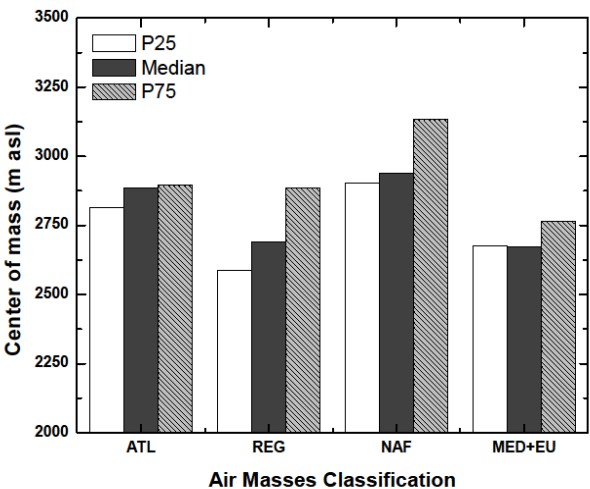

**Figure 7: Bar-plot of the center of mass (m a.s.l.) of the 25th percentile (P25), median and 75th percentile (P75) profiles, separated as a function of air mass (ATL = Atlantic, REG = Regional, MED+EU = Mediterranean and European, NAF = North African).**

