# Peer review of "Retrieval of aerosol properties from ceilometer and photometer measurements: long-term evaluation with in-situ data and statistical analysis at Montsec (southern Pyrenees)"

_Atmospheric Measurement Techniques, 2018_

## Referee Comment (RC1) · Barry Gross (Referee) · 1 Mar 2019

This paper entitled "Retrieval of aerosol properties from ceilometer and photometer measurements: long-term evaluation with in-situ data and statistical analysis at Montsec (southern Pyrenees)" provides a very detailed statistical assessment of a fairly recent algorithmic approach entitled GRASP ( Generalized Retrieval of Aerosol and Surface Properties) to optimize the vertical retrieval of aerosol properties by merging vertical profile data with constraining column data from a sun/sky photometer.

[Figure]

The novelty of the statistical comparison is that it is done using insitu data obtained from the Montsec Observatory which is ∼ 750 meters above the ceilometer profiling instrument which allows the authors to study the retrieval performance above the critical overlap region which even with correction generally leads to significant biases that would be enhanced by the multi-instrument retrieval. Another novelty of the paper is the long duration of the study (3 years) which allows the authors to study the effects of different meteorology and aerosol sources on the results. Based on the authors literature background, this is a significant improvement over existing algorithm retrieval validations which were limited to short duration air craft campaigns.

Besides quantifying the retrieval characteristics and identifying conditions in which the retrieval performances is degraded (fine mode dominated), the authors are able to build a very useful profile climatology of vertical aerosol properties filtered by climatology (RH) and source locations

In summary, the paper illustrates convincingly the usefulness of the GRASP algorithm in optimally analyzing combined Ceilometer / Sky Radiometer data. The fact that a much cheaper ceilometer is used instead of more costly and sophisticated Lidars is very useful since it opens up the possibility of developing a much larger network of such instrument sites.

Suggestions:

One suggestion that comes to mind would be to illustrate the importance of the radiometer (i.e radiance constraints) on the retrieval properties. In particular, prior algorithms that combine lidar (or ceilometers) with only total column Aerosol Optical Depth at multiple wavelengths from a sunphotometer has been used to retrieve vertical aerosol profiles. The sun photometer AOD measurements may provide an even cheaper alternative to an instrument site. Therefore, it would be very helpful to the community if some comparison between the different algorithms could be made to see how much improvement there is in the AERONET Radiometer data as compared to

just sunphotometer AOD constraints on this very nice and rich data set.

---

## Referee Comment (RC2) · Anonymous Referee #1 · 8 Mar 2019

This paper addresses an evaluation of the aerosol property profiles retrieved from GRASP algorithm and which uses as inputs ceilometer and sun-photometer (SPM) measurements versus in-situ measurements. The work presents different relevant aspects that show its importance and novelty. This is the first time that GRASP algorithm using as inputs ceilometer and SPM measurements (GRASPpac) has been evaluated in a long-term comparison. This new approach (GRASPpac) presents big advantages since these two instruments can be operated in a continuous and almost unattended way and its use has been expanded by networks providing much more global cover-

age. However, before this approach is widely used its products need to be validated as is done in this work.

In addition the work have dealt with the complexity of comparing different techniques (remote and in-situ) which also cover different ranges in the Earth-atmosphere system (surface and almost full troposphere). The results presented here show a good agreement between the optical properties from techniques and larger discrepancies in the volume size distribution when fine particles are dominant.

So after these comments I conclude that the paper is very interesting, well written and show the capability of GRASPpac approach to retrieve vertical information of aerosol properties based on this long-term study. I consider that this work is appropriated for Atmospheric Measurement Techniques and it should be published after some minor corrections:

- About the comparison: some explanations should be given about how the in-situ measurements and the GRASP profiles are compared. How many points from the lowest part of the profiles do you take? Do you average those points? What is the altitude range that they represent? The lowest part of the remote sensing profiles are always more problematic due to the incomplete overlap of the ceilometer (even if it is corrected with the overlap function provided by the manufacturer). So I consider that is important to discuss these points in the manuscript.

- Looking at the histograms presented in Figure 2 I have the impression that the distributions of the relative differences are bounded to a certain positive value, how do you explain that there are no observations with discrepancies larger than +1%?

- Page 2, line 30: it should be also indicated that ceilometer provides continuous measurements, in contrast with most of the "more sophisticated" lidar systems.

- Page 4, line 22: I wonder why you use Aeronet data level 1.5. For this long-term study level 2 data (quality assured) should be available.

- Page 4, line 27: authors indicate that the ceilometer is located 760 meters downslope of MSA station, but what is the different in altitude between both measurement sites? In addition some lines below authors indicate that the overlap is higher than 85% at 760 m (same number), is it a typo or just a coincidence?

- Page 6, line 20: I understand that in Figure 3 authors compare the lowest part of GRASP profiles, so please indicate it.

- Page 7, line 16: delete the parenthesis: ". . . before)."

- Page 11, line 1: Ceilometers can be considered automated lidars, so just mention the first one. "the use of automated lidars for the determination . . ..".

- Page 11, line 5: Correct the next typo: (ceilometer and lidar have to be in singular) ". . . ceilometer and lidar networks . . ."

---

## Referee Comment (RC3) · Anonymous Referee #3 · 14 Mar 2019

The paper is an interesting contribution to aerosol remote sensing as it highlights the potential of automated ceilometers being available in large numbers (networks operated by national weather services). The limitation in characterizing aerosols is caused by the low power and the single wavelength compared to advanced lidar systems. However, the very good spatial and temporal coverage is a big advantage (unattended continuous operation). To overcome the above mentioned limitation the joint exploitation with photometer measurements is proposed (by means of $GRASP_{pac}$) – a validation of this approach is provided in this manuscript. Moreover, long-term observations of

aerosol profiles (in terms of extinction and derived by the novel GRASP-approach) are presented.

The paper is clearly structured, well written and relevant. There are several promising applications: the benefit of successful retrievals of aerosol profiles (backscatter or extinction) with high temporal resolution (as described in this paper) could be enormous for model validation: up to now validation is mainly confined to surface values (PM10, PM2.5) or columnar values (AOD), e.g. in the framework of AQMEII. Consequently, I recommend publication in AMT. Only minor changes are suggested. Most of them are just details in wording – nevertheless a few clarifications to avoid possible misunderstandings are strongly recommended.

In the following page and lines are given in square brackets

- [2,23] "Aerosol Optical Depth". I would suggest to use lower case letters.

- [2,24] "range corrected signal (RCS) lidar values..." → "range corrected lidar signal (RCS)..."

- [2,29] "...the GRASP algorithm is a significant advance...". Maybe change "is" to "can be", as the (positive) result of this paper is not yet known when reading the introduction.

- [2,31] I would be cautious with the word "worldwide" (see also the abstract): There are quite different ceilometers in operation and it is not yet clear whether the conclusions of this paper (found for one CHM15k) can be transferred to other systems: Many ceilometers have a lower pulse energy (consequently a limited measurement range) than the CHM15k (e.g., CL31, CHM8k), are influenced by water vapor absorption (CL31, CL51, CHM8k) and dense networks do not exist in all parts of the world. It is not unlikely that $GRASP_{pac}$ can be applied to some other systems as well, but to my knowledge this has not yet been demonstrated.

This topic should briefly be discussed (maybe in the conclusions). There are certainly publications on these issues that might be useful. See also comment on [5,14].

- [3,1] I agree that the methodology presented in this paper can be a step forward, however, there are other options for quantitative aerosol remote sensing (including night time measurements). In the last years several publications have demonstrated that ceilometers can provide the particle backscatter coefficient (under certain atmospheric conditions, see e.g. Wiegner and Geiß, 2012). They use a different approach than Titos et al. by using calibrated ceilometer data and get an uncertainty in the order of 10% for the backscatter coefficient. How does this compare to GRASP$_{pac}$ (same order of magnitude?)? See also comment on [8,19].

- [4,2] Whenever "backscattering" is used in the paper make clear whether backscattering in the "lidar sense" is meant (i.e. scattering under 180°; in m$^{-1}$ sr$^{-1}$) or "hemispheric backscattering" (in m$^{-1}$) as measured by the nephelometer. And explain how the comparison is made: From Section 4.1.1 I understand that it is assumed that scattering into the backward hemisphere is the same for all directions. Then, from the integral (scattering angles $90° \leq \theta \leq 171°$, extrapolated to scattering angles $90° \leq \theta \leq 180°$) the authors calculate a mean backscatter coefficient (which – under these assumptions – can also be applied to 180°) and compare this value with the GRASP$_{pac}$-retrieval. As scattering under 180° is typically larger than scattering under smaller angles, I would expect an overestimation by GRASP (see also discussion in [6,6ff]). That was indeed found. As this is one of the main topics of this paper, the authors should be very clear – this might induce an extension of the discussion.

- [4,5] What is "lpm"? Should be "l min$^{-1}$"?

- [4,32] "...using the manufacturer's overlap function." This seems to be for information only, with no consequences for the retrieval as the vertical difference between the ceilometer site and the observatory is 760 m (correct? Or what does "downslope" mean? If the vertical difference is smaller, the overlap issue should be discussed in more detail. On the other hand a horizontal distance of 2.5 km is mentioned.). Or is there another reason for mentioning this? Please avoid confusion of the reader.

• [5,12] Please explain what "normalized" ceilometer RCS means?

• [5,14] "corresponds to the MSA altitude..." So RCS between 760 m and 7760 m (above sea level, distance from the ceilometer $<$ 7 km) are considered in the inversion? Here I would expect a comment on the measurement range of the CHM15k: data are available up to 15 km ([4,27]) but the range that can be exploited is smaller. In the framework of the CeiLinEx2015 campaign it was shown that in the free troposphere the CHM15k signals are quite noisy (the maximum range of Vaisala- and Campbell-ceilometers is even smaller; this may influence the "worldwide"-discussion from above as well). How does this affect the GRASP$_{pac}$-retrieval? Is the maximum range (7000 m) reduced, but keeping the 60 levels?

• [5,20] The authors mention that the retrieved $r_{eff}$ is height-independent, but never use $r_{eff}$ in the paper. So it is recommended to mention the volume concentration $V$ instead (or in addition).

• [5,21] "backscattering": here "under $180°$"?

• [5,22] Please add a short comment on the accuracy of the retrieved aerosol parameters that are used in Section 4 (see also comment [3,1]). This is an important/mandatory information.

• [5,26] "backscattering": here "hemispheric"?

- [5,30] Give an equation/definition of "comparison" and "relative difference" (see also [6,2] and figures): "GRASP minus in-situ" or "in-situ minus GRASP"? Divided by the "mean of in-situ"?

- [6,7] Is the angular range with respect to the backscatter configuration correct? It should be $90°$ instead of $10°$ here (cf. Müller et al., 2011a)?

- [6,12] "tends to overestimate all the studied...". Is this statement trivial?

- [6,22] "...there is a linear trend between scattering and extinction coefficients...": What does this mean? If scattering and extinction are the same, the single scattering albedo $\omega$ is $\omega = 1$. This is more or less the case under clean and turbid conditions according to Fig. 3a. If scattering and extinction show a linear dependence, the single scattering albedo is constant. The fact that in general large extinction coefficients correspond to large scattering coefficients is not surprising ($\omega$ is typically between say 0.8 and 1).

- [8,17] "Qualitatively speaking, the volume concentration...". I don't understand the reason for this sentence? The seasonal cycle of the volume size distribution is not shown in the paper. Is this sentence included because a similar behavior is plausible? Or because this is known from literature? Or is it an intrinsic feature of the GRASP-methodology?

- [8,19] A comment on the limitation to daytime: This is caused by the combination with the sun photometer. From the ceilometer's perspective the determination of backscatter coefficients can be provided during night time as well (likely even better) provided that a calibration is possible, see comment on [3,1]. The extinction coefficient can be estimated if the lidar ratio can be estimated (of course, subject to – maybe significant – errors). In Titos et al.'s paper the transformation from RCS to physical quantities is provided by using the photometer data

as constraint; this is conceptually superior (at the expense of the two measuring systems required).

- [8,23] above → larger than

- [10,1] Explain "center of mass"; e.g. by citing Mona et al. (2006) or alternatively (if you want to avoid another self-citation) Binietoglou et al. (2015).

- [10,4] Here and in Fig. 7, the center of mass is given with respect to sea level. The numbers are correct but may lead to misunderstandings as the reader would intuitively expect much lower values (main contribution is almost always from the mixing layer). Indeed, values of 1.0–1.5 km can be found from Fig. 7 when considering heights above ground. So I recommend to add the corresponding values in brackets, at least in one or two cases.

- [10,28] "Similar seasonal behavior...": This is not a result of this study, it is only a message from a paper by Pandolfi et al. (2014). This should be made clear.

- [11,12] Here, Illingworth et al. (2018) can be cited.

- [Fig. 5] delete "light" in the caption; also in Fig. 6.

Suggested references:

Binietoglou, I., Basart, S., Alados-Arboledas, L., Amiridis, V., Argyrouli, A., Baars, H., Baldasano, J. M., Balis, D., Belegante, L., Bravo-Aranda, J. A., Burlizzi, P., Carrasco, V., Chaikovsky, A., Comerón, A., D'Amico, G., Filioglou, M., Granados-Munoz, M. J., Guerrero-Rascado, J. L., Ilic, L., Kokkalis, P., Maurizi, A., Mona, L., Monti, F., Munoz-Porcar, C., Nicolae, D., Papayannis, A., Pappalardo, G., Pejanovic, G., Pereira, S. N., Perrone, M. R., Pietruczuk, A., Posyniak, M., Rocadenbosch, F., Rodriguez-Gómez, A., Sicard, M., Siomos, N., Szkop, A., Terradellas, E., Tsekeri, A., Vukovic, A., Wandinger, U., and Wagner, J.: A methodology for investigating dust model performance using

synergistic EARLINET/AERONET dust concentration retrievals, Atmos. Meas. Tech., 8, 3577-3600, https://doi.org/10.5194/amt-8-3577-2015, 2015.

Mona, L., A. Amodeo, M. Pandolfi, and G. Pappalardo (2006), Saharan dust intrusions in the Mediterranean area: Three years of Raman lidar measurements, J. Geophys. Res., 111, D16203, doi:10.1029/2005JD006569.

Illingworth, A., Cimini, D., Haefele, A., Haeffelin, M., Hervo, M., Kotthaus, S., Löhnert, U., Martinet, P., Mattis, I., O'Connor, E., and Potthast, R.: How can Existing Ground-Based Profiling Instruments Improve European Weather Forecasts?, Bull. Amer. Meteorol. Soc., https://doi.org/10.1175/BAMS-D-17-0231.1, 2018.

Wiegner, M. and Geiß, A.: Aerosol profiling with the Jenoptik ceilometer CHM15kx, Atmos. Meas. Tech., 5, 1953-1964, https://doi.org/10.5194/amt-5-1953-2012, 2012.

---

## Author Comment (AC1) · 7 May 2019

We thank all the reviewers for their comments and suggestions that have helped to improve the quality of the manuscript. A point by point response to Prof. Gross is included below. Changes in the manuscript are noted between quotation marks. The new version of the manuscript with the changes tracked is included in a separate file.

Barry Gross (Referee) Comment: This paper entitled "Retrieval of aerosol properties from ceilometer and photometer measurements: long-term evaluation with in-situ data

and statistical analysis at Montsec (southern Pyrenees)" provides a very detailed statistical assessment of a fairly recent algorithmic approach entitled GRASP ( Generalized Retrieval of Aerosol and Surface Properties) to optimize the vertical retrieval of aerosol properties by merging vertical profile data with constraining column data from a sun/sky photometer. The novelty of the statistical comparison is that it is done using insitu data obtained from the Montsec Observatory which is _ 750 meters above the ceilometer profiling instrument which allows the authors to study the retrieval performance above the critical overlap region which even with correction generally leads to significant biases that would be enhanced by the multi-instrument retrieval. Another novelty of the paper is the long duration of the study (3 years) which allows the authors to study the effects of different meteorology and aerosol sources on the results. Based on the authors literature background, this is a significant improvement over existing algorithm retrieval validations which were limited to short duration air craft campaigns. Besides quantifying the retrieval characteristics and identifying conditions in which the retrieval performances is degraded (fine mode dominated), the authors are able to build a very useful profile climatology of vertical aerosol properties filtered by climatology (RH) and source locations In summary, the paper illustrates convincingly the usefulness of the GRASP algorithm in optimally analyzing combined Ceilometer / Sky Radiometer data. The fact that a much cheaper ceilometer is used instead of more costly and sophisticated Lidars is very useful since it opens up the possibility of developing a much larger network of such instrument sites. Suggestions: One suggestion that comes to mind would be to illustrate the importance of the radiometer (i.e radiance constraints) on the retrieval properties. In particular, prior algorithms that combine lidar (or ceilometers) with only total column Aerosol Optical Depth at multiple wavelengths from a sunphotometer has been used to retrieve vertical aerosol profiles. The sun photometer AOD measurements may provide an even cheaper alternative to an instrument site. Therefore, it would be very helpful to the community if some comparison between the different algorithms could be made to see how much improvement there is in the AERONET Radiometer data as compared to just sunphotometer AOD constraints on this very nice

and rich data set.

Answer: We thank Prof. Gross for his positive comments on this manuscript. The GRASP retrieval using only AOD and ceilometer signal is an interesting suggestion that looks very promising since sky radiance measurements are scarcer than AOD measurements. In line with this comment, Torres et al. (2017) explored the use of multi-wavelength AOD measurements to retrieve column integrated size aerosol properties. The versatility of GRASP algorithm is promoting the use of different set of data as inputs such as sky radiances from sky cameras combined with lunar photometers (Román et al., 2017) or scattering measurements from polar nephelometers to retrieve aerosol optical and microphysical properties (Espinosa et al., 2017). In this study we focus on the evaluation of the inversion scheme proposed by Román et al. (2018) that uses sun/sky photometer and includes the RCS from ceilometer as novelty. We are currently exploring other inversion strategies and we will consider the use of only AOD as suggested by the reviewer. However, we feel that this would be out of the scope of the present manuscript in which the main objective is to evaluate the GRASPpac inversion strategy.

References: Espinosa, W.R., Remer, L.A., Dubovik, O., Ziemba, L., Beyersdorf, A., Orozco, D., Schuster, G., Lapyonok, T., Fuertes, D., and Martins, J.V.: Retrievals of aerosol optical and microphysical properties from imaging polar nephelometer scattering measurements, Atmos. Meas. Tech., 10, 811–824, https://doi.org/10.5194/amt-10-811-2017, 2017.

Román, R., Torres, B., Fuertes, D., Cachorro, V.E., Dubovik, O., Toledano, C., Cazorla, A., Barreto, A., Bosch, J.L., Lapyonok, T., González, R., Goloub, P., Perrone, M.R., Olmo, F.J., de Frutos, A., and Alados-Arboledas, L.: Remote sensing of lunar aureole with a sky camera: adding information in the nocturnal retrieval of aerosol properties with GRASP code, Remote Sens. Environ., 196, 238–252. https://doi.org/10.1016/j.rse.2017.05.013, 2017.

Román, R., Benavent-Oltra, J.A., Casquero-Vera, J.A., Lopatin, A., Cazorla, A., Lyamani, H., Denjean, C., Fuertes, D., Pérez-Ramírez, D., Torres, B., Toledano, C., dubovik, O., Cachorro, V.E., de Frutos, A.M., Olmo, F.J., and Alados-Arboledas, L.: Retrieval of aerosol profiles combining sunphotometer and ceilometer measurements in GRASP code, Atmos. Res., 204, 161–177, https://doi.org/10.1016/j.atmosres.2018.01.021, 2018.

Torres, B., Dubovik, O., Fuertes, D., Lapyonok, T., Toledano, C., Schuster, G. L., Goloub, P., Blarel, L., Barreto, A., Mallet, M., Tanré, D.: Advanced characterization of aerosol properties from measurements of spectral optical depth using the GRASP algorithm, Atmos. Meas. Tech., 10, 3743-3781, https://doi.org/10.5194/amt-10-3743-2017, 2017.

Please also note the supplement to this comment:
https://www.atmos-meas-tech-discuss.net/amt-2018-431/amt-2018-431-AC1-supplement.pdf
* * *
[Figure]

**Supplement:**

[revised manuscript text omitted]

---

## Author Comment (AC2) · 7 May 2019

We thank all the reviewers for their comments and suggestions that have helped to improve the quality of the manuscript. A point by point response to Reviewer1's comments is included below. Changes in the manuscript are noted between quotation marks. The new version of the manuscript with the changes tracked is included in a separate file.

Anonymous Referee #1 Comment: This paper addresses an evaluation of the aerosol

property profiles retrieved from GRASP algorithm and which uses as inputs ceilometer and sun-photometer (SPM) measurements versus in-situ measurements. The work presents different relevant aspects that show its importance and novelty. This is the first time that GRASP algorithm using as inputs ceilometer and SPM measurements (GRASPpac) has been evaluated in a long-term comparison. This new approach (GRASPpac) presents big advantages since these two instruments can be operated in a continuous and almost unattended way and its use has been expanded by networks providing much more global coverage. However, before this approach is widely used its products need to be validated as is done in this work. In addition the work have dealt with the complexity of comparing different techniques (remote and in-situ) which also cover different ranges in the Earth-atmosphere system (surface and almost full troposphere). The results presented here show a good agreement between the optical properties from techniques and larger discrepancies in the volume size distribution when fine particles are dominant. So after these comments I conclude that the paper is very interesting, well written and show the capability of GRASPpac approach to retrieve vertical information of aerosol properties based on this long-term study. I consider that this work is appropriated for Atmospheric Measurement Techniques and it should be published after some minor corrections: About the comparison: some explanations should be given about how the in-situ measurements and the GRASP profiles are compared. How many points from the lowest part of the profiles do you take? Do you average those points? What is the altitude range that they represent? The lowest part of the remote sensing profiles are always more problematic due to the incomplete overlap of the ceilometer (even if it is corrected with the overlap function provided by the manufacturer). So I consider that is important to discuss these points in the manuscript.

Answer: We agree with the reviewer that this explanation is missing in the manuscript. Since the sun/sky photometer is located at the Montsec observatory (1570 m a.s.l.) and the ceilometer is located at 800 m a.s.l., the combination of RCS from the ceilometer and sun/sky photometer measurements in the GRASP algorithm is only possible from

1570 m a.s.l. and onwards. The ceilometer RCS used as input in GRASP is normalized and averaged at 60 log-spaced points, being the first point (1/60) equal to 1570 m a.s.l. Therefore, the GRASP-derived profiles start at 1570 m a.s.l. The comparison with the in-situ measurements is therefore made at 1570 m a.s.l. and consequently the comparison is only representative of this altitude (we cannot assure that the results of the comparison are similar at higher or lower altitudes since the in-situ measurements are limited to a single point). Also, we agree with the reviewer that the lower part of the remote sensing profiles are always more problematic due to incomplete overlap issues, and this might affect the comparison. However, since the overlap of the telescope and laser beam is greater than 85% at the height of the MSA station (1570 m a.s.l.), this effect is expected to be low. We have discussed these points with more detail in the revised version of the manuscript.

Section 2.4.: It has been modified as follows: "The ceilometer is located at 800 m a.s.l., at the Center for the Observation of the Universe (COU, http://www.parcastronomic.cat/). The horizontal distance between the ceilometer and the MSA station is less than 2.5 km."

Section 4.1.1.: We have included: "The comparison has been performed at 1570 m a.s.l., where the in-situ instrumentation is located and coinciding with the first height of the GRASPpac retrievals. Therefore, the following results and associated discussion on the comparison between GRASPpac and in-situ measurements refer exclusively to this height."

Section 2.4: we have modified it as follows: "The RCS profiles provided by the instrument are overlap-corrected using the manufacturer's overlap function. In addition, according to this function, the overlap of the telescope and the laser beam is greater than 85% at around 770 m a.g.l. Thus, the effect of the overlap at the height of the MSA observatory (1570 m a.s.l.) is expected to be low."

Section 3: we have included: "As mentioned before, the RCS profiles provided by the

ceilometer are overlap-corrected and, according to the manufacturer's overlap function, the overlap of the telescope and the laser beam is greater than 85% at the MSA altitude (770 m above the ceilometer). Thus, the effect of the overlap in the GRASPpac retrievals is expected to be low, since the ceilometer RCS below 1570 m a.s.l. is not used here as input in GRASPpac."

Comment: Looking at the histograms presented in Figure 2 I have the impression that the distributions of the relative differences are bounded to a certain positive value, how do you explain that there are no observations with discrepancies larger than +1%?

Answer: Histograms presented in Figure 2 show the relative difference between in-situ measurements and GRASP derived optical variables, taking the in-situ measurements as reference (calculated as the difference between in-situ and GRASPpac values divided by the in-situ value). We agree with the reviewer that the relative difference seems biases to lower positive relative errors. To avoid this, we have replaced relative difference by absolute difference (in-situ minus GRASPpac values) in Figure 2 and Figure 4, as well as the related discussion in the manuscript. We think that these new figures provide a better idea on the difference between GRASPpac and in-situ measurements.

Comment: Page 2, line 30: it should be also indicated that ceilometer provides continuous measurements, in contrast with most of the "more sophisticated" lidar systems.

Answer: Done.

Comment: Page 4, line 22: I wonder why you use Aeronet data level 1.5. For this long-term study level 2 data (quality assured) should be available.

Answer: We use Level 1.5 data because typically Level 2.0 data is not available after one year, when the post-calibration of the photometer is done. GRASPpac has the potential to derive aerosol properties in near real time if it is feed with Level 1.5 Aeronet data, but not if Level 2.0 data is used. Hence, we think that the study of the

performance of GRASPpac feed with Level 1.5 data is more interesting than with Level 2.0 due to the near real time implications.

Please also note the supplement to this comment:
https://www.atmos-meas-tech-discuss.net/amt-2018-431/amt-2018-431-AC2-supplement.pdf

—————————————————————

---

## Author Comment (AC3) · 7 May 2019

We thank all the reviewers for their comments and suggestions that have helped to improve the quality of the manuscript. A point by point response to the comments of Reviewer#3 is included below. Changes in the manuscript are noted between quotation marks. The new version of the manuscript with the changes tracked is included in a separate file.

Anonymous Referee #3

[Figure]

Comment: The paper is an interesting contribution to aerosol remote sensing as it highlights the potential of automated ceilometers being available in large numbers (networks operated by national weather services). The limitation in characterizing aerosols is caused by the low power and the single wavelength compared to advanced lidar systems. However, the very good spatial and temporal coverage is a big advantage (unattended continuous operation). To overcome the above mentioned limitation the joint exploitation with photometer measurements is proposed (by means of GRASP-pac) – a validation of this approach is provided in this manuscript. Moreover, long-term observations of aerosol profiles (in terms of extinction and derived by the novel GRASP-approach) are presented. The paper is clearly structured, well written and relevant. There are several promising applications: the benefit of successful retrievals of aerosol profiles (backscatter or extinction) with high temporal resolution (as described in this paper) could be enormous for model validation: up to now validation is mainly confined to surface values (PM10, PM2.5) or columnar values (AOD), e.g. in the framework of AQMEII. Consequently, I recommend publication in AMT. Only minor changes are suggested. Most of them are just details in wording – nevertheless a few clarifications to avoid possible misunderstandings are strongly recommended. In the following page and lines are given in square brackets

Comment: [2,23] "Aerosol Optical Depth". I would suggest to use lower case letters.

Answer: Done

Comment: [2,24] "range corrected signal (RCS) lidar values..." -> "range corrected lidar signal (RCS)..."

Answer: Done

Comment: [2,29] "...the GRASP algorithm is a significant advance...". Maybe change "is" to "can be", as the (positive) result of this paper is not yet known when reading the introduction.

Answer: Done

Comment: [2,31] I would be cautious with the word "worldwide" (see also the abstract): There are quite different ceilometers in operation and it is not yet clear whether the conclusions of this paper (found for one CHM15k) can be transferred to other systems: Many ceilometers have a lower pulse energy (consequently a limited measurement range) than the CHM15k (e.g., CL31, CHM8k), are influenced by water vapor absorption (CL31, CL51, CHM8k) and dense networks do not exist in all parts of the world. It is not unlikely that GRASPpac can be applied to some other systems as well, but to my knowledge this has not yet been demonstrated. This topic should briefly be discussed (maybe in the conclusions). There are certainly publications on these issues that might be useful. See also comment on [5,14].

Answer: We agree with the reviewer. We have omitted the word "worldwide" in the manuscript. Also, we have included in the conclusion section a brief discussion on the applicability of GRASPpac to other ceilometers with lower capabilities than the one used here.

"Nevertheless, it is important to bear in mind that the results presented in this study are limited to day time and low cloudiness conditions due to the need of simultaneous sun/sky photometer measurements. Also, further studies investigating the performance of the application of GRASPpac to ceilometers and automatic lidars with different characteristics (i.e. wavelength of operation, pulse energy) than the one used in this study are needed to maximize its potential application. With this in mind, the implementation of GRASPpac in the frame of measurement networks will contribute to enhance the representativeness of aerosol vertical distribution providing useful information for satellite and models evaluation, and contributing to the objectives of several international initiatives (Illingworth et al., 2018) such us the EU COST Action TOPROF (Towards operational ground-based profiling with ceilometers, Doppler lidars and microwave radiometers for improving weather forecasts) or the E-PROFILE program of the European Meteorological Services Network."

Comment: [3,1] I agree that the methodology presented in this paper can be a step forward, however, there are other options for quantitative aerosol remote sensing (including night time measurements). In the last years several publications have demonstrated that ceilometers can provide the particle backscatter coefficient (under certain atmospheric conditions, see e.g. Wiegner and Geiß, 2012). They use a different approach than Titos et al. by using calibrated ceilometer data and get an uncertainty in the order of 10% for the backscatter coefficient. How does this compare to GRASPpac (same order of magnitude?)? See also comment on [8,19].

Answer: We certainly agree with the reviewer and believe that a discussion on previous studies retrieving quantitative aerosol optical information (i.e. backscatter coefficients) from ceilometers' RCS signal is missing in the manuscript. We have included this discussion in the introduction and conclusion sections, with appropriate references.

"The use of ceilometer measurements in the GRASP algorithm can be a significant advance towards a better representation of aerosol properties with vertical resolution since ceilometers are cheaper, require less supervision, provide continuous measurements and are more extensively distributed compared to more sophisticated lidar systems (Wiegner et al., 2014; Cazorla et al., 2017; Dionisi et al., 2018). However, the main drawback of this approach is that sun/sky photometer measurements are only available during day time and under low cloudiness conditions. Other methodologies, such as the absolute calibration of the ceilometer (Wiegner and Geiß, 2012), are able to overcome this issue and provide quantitative backscatter profiles during day and night time. Quantitative ceilometer profiles could be used for evaluating dust forecast models (Tsekeri et al., 2017) such as the BSC-DREAM8b, as input to radiative transfer models (Granados-Muñoz et al., 2019) or can be assimilated in global models (Chen et al., 2018). This application represents a step-forward in the classical use of ceilometers that were originally developed for cloud base detection (e.g., Martucci et al., 2010; Wiegner et al., 2014)."

Concerning the uncertainty of GRASPpac retrieved backscatter coefficient, Román et

al. (2018) estimated a mean uncertainty of 31% analyzing synthetic data under various scenarios. This estimated uncertainty is lower for the volume concentration and the extinction coefficient (21%). We have included the uncertainties in the revised version of the manuscript, and a comment on the lower uncertainty estimated by Wiegner and Geiß (2012) for their calculation.

"Román et al. (2018) estimated the GRASPpac uncertainty analyzing synthetic data under various scenarios. According to these authors, the backscatter coefficient mean uncertainty is 31%, and 21% for the extinction coefficient and the volume concentration. The uncertainty in the backscatter profiles retrieved with GRASPpac is higher than the estimated uncertainty by Wiegner and Geis (2012) for the absolute calibration method (10%)."

Comment: [4,2] Whenever "backscattering" is used in the paper make clear whether backscattering in the "lidar sense" is meant (i.e. scattering under 180-1; in m-1 sr-1) or "hemispheric backscattering" (in m-1) as measured by the nephelometer. And explain how the comparison is made: From Section 4.1.1 I understand that it is assumed that scattering into the backward hemisphere is the same for all directions. Then, from the integral (scattering angles $90 \leq \theta \leq 171°$, extrapolated to scattering angles $90 \leq \theta \leq 180°$) the authors calculate a mean backscatter coefficient (which – under these assumptions – can also be applied to 180°) and compare this value with the GRASPpac-retrieval. As scattering under 180° is typically larger than scattering under smaller angles, I would expect an overestimation by GRASP (see also discussion in [6,6ff]). That was indeed found. As this is one of the main topics of this paper, the authors should be very clear – this might induce an extension of the discussion.

Answer: That is correct, we are assuming that the scattering into the backward hemisphere is the same in all directions. In section 3 we have included the following paragraph explaining how the comparison has been done:

"Since the in-situ measurements and GRASPpac retrievals provide different informa-

tion with respect to the aerosol backward scattering properties (hemispheric backscattering versus backscatter at 180°) the direct comparison between both techniques is not possible. To have a sense of the performance of the GRASPpac backscatter retrieval, for the comparison we have assumed that the scattering into de backward hemisphere is the same in all directions. Therefore, we have extrapolated the backscatter at 180° to the angular range 90-180° in order to make it comparable with the backscattering coefficient measured with the nephelometer. This assumption constitutes an additional source of error since the actual angular scattering distribution is not known and typically backscatter at 180° is larger than at smaller angles."

Comment: [4,5] What is "lpm"? Should be "l min-1"?

Answer: Yes, it is liters per minute. We have changed the notation.

Comment: [4,32] "...using the manufacturer's overlap function." This seems to be for information only, with no consequences for the retrieval as the vertical difference between the ceilometer site and the observatory is 760 m (correct? Or what does "downslope" mean? If the vertical difference is smaller, the overlap issue should be discussed in more detail. On the other hand a horizontal distance of 2.5 km is mentioned.). Or is there another reason for mentioning this? Please avoid confusion of the reader.

Answer: The vertical distance between the ceilometer and the Montsec observatory is 770 m (there was a typo in the original version, and the height difference is 770 instead of 760 meters). The Montsec observatory where the in-situ instruments and the sun/sky photometer are located is at 1570 m a.s.l. and the ceilometer is located at 800 m a.s.l. We have clarified this point in the revised version of the manuscript, avoiding the term downslope that was misleading. On the other hand, the horizontal distance between the two sites is 2.5 km. We have included this information because a short horizontal distance between the two set of instruments will likely guarantee that both are sampling the same air mass, while for longer horizontal distances the probability of different air masses affecting each site increases. The following information has been

included / modified in the revised version of the manuscript:

Section 2.4.: It has been modified as follows: "The ceilometer is located at 800 m a.s.l., at the Center for the Observation of the Universe (COU, http://www.parcastronomic.cat/). The horizontal distance between the ceilometer and the MSA station is less than 2.5 km."

Section 4.1.1.: We have included: "The comparison has been performed at 1570 m a.s.l., where the in-situ instrumentation is located and coinciding with the first height of the GRASPpac retrievals. Therefore, the following results and associated discussion on the comparison between GRASPpac and in-situ measurements refer exclusively to this height. "

Section 2.4: we have modified it as follows: "The RCS profiles provided by the instrument are overlap-corrected using the manufacturer's overlap function. In addition, according to this function, the overlap of the telescope and the laser beam is greater than 85% at around 770 m a.g.l. Thus, the effect of the overlap at the height of the MSA observatory (1570 m a.s.l.) is expected to be low."

Section 3: we have included: "As mentioned before, the RCS profiles provided by the ceilometer are overlap-corrected and, according to the manufacturer overlap function, the overlap of the telescope and the laser beam is greater than 85% at the MSA altitude (770 m above the ceilometer). Thus, the effect of the overlap in the GRASPpac retrievals is expected to be low, since the ceilometer RCS below 1570 m a.s.l. is not used here as input in GRASPpac."

Comment: [5,12] Please explain what "normalized" ceilometer RCS means?

Answer: The RCS is normalized at 60 log-spaced bins at different heights, as in Lopatin et al. (2013), being the minimum of these heights (zmin) the MSA altitude (1570 m a.s.l.). The maximum height (zmax) selected for the 60 log-spaced bins is 7000 m above MSA since aerosol layers are rarely detected above this height and the ceilometer signal is usually too noisy at higher altitudes due to the low power of the ceilometer's laser (please see our next comment about the maximum height). The RCS at these 60 log-spaced bins is normalized by dividing the average of RCS in each logarithmic height interval by the integrated RCS between zmin and zmax. Therefore, the normalized RCS provides a value of 1 when it is integrated. This is done because GRASP algorithm uses normalized signals as input. Further details can be found in Román et al. (2018). For clarification we have included the following information in section 3:

"The minimum height of these 60 values corresponds to the MSA altitude. The maximum height selected for the 60 log-spaced bins is 7000 m above MSA, since aerosol layers are rarely detected above this height and the ceilometer signal is usually too noisy at higher altitudes due to the low power of the ceilometer's laser. The RCS at these 60 log-spaced bins is averaged and then normalized by dividing each value by the integrated RCS between the minimum and maximum heights."

Comment: [5,14] "corresponds to the MSA altitude..." So RCS between 760 m and 7760 m (above sea level, distance from the ceilometer < 7 km) are considered in the inversion? Here I would expect a comment on the measurement range of the CHM15k: data are available up to 15 km ([4,27]) but the range that can be exploited is smaller. In the framework of the CeiLinEx2015 campaign it was shown that in the free troposphere the CHM15k signals are quite noisy (the maximum range of Vaisala- and Campbell-ceilometers is even smaller; this may influence the "worldwide"-discussion from above as well). How does this affect the GRASPpac-retrieval? Is the maximum range (7000 m) reduced, but keeping the 60 levels?

Answer: We agree with the reviewer, the quality of RCS from ceilometers decreases with height. The minimum height of the 60 log-spaced height values corresponds to the MSA observatory altitude (1570 m a.s.l.) and the maximum height has been assumed up to 7000 m above the observatory altitude, therefore up to 8570 m a.s.l. This altitude corresponds with a height of 7770 meters above the ceilometer, therefore, although RCS data from the ceilometer is available up to 15 km, only ∼8 km are considered.

This value is slightly larger than that considered by Román et al. (2018) in order to reach a height of around 7000 m above the observatory. In those cases which the RCS is too noisy, the normalized RCS values at high altitude are negative and noisy, and hence, following the criteria of Román et al. (2018), an iterative process is applied to the RCS values where the maximum height (7000 m above the observatory) is iteratively reduced every 100 meters until all the RCS values are positive. This process means that the maximum altitude reached with GRASPpac is not always 7000 m above the observatory, since it could be lower. The number of height-levels is always 60 independently of the maximum height attained.

Comment: [5,20] The authors mention that the retrieved re is height-independent, but never use re in the paper. So it is recommended to mention the volume concentration V instead (or in addition).

Answer: We have clarified this point in the revised version of the manuscript. It is important to keep in mind that GRASPpac assumes that intensive properties (such as the effective radius) does not change with height while it considers changes with height in the extensive aerosol properties (like volume concentration).

"Since ceilometer measurements are limited to a single wavelength, it is not possible to vertically differentiate between aerosol modes/types and therefore vertical profiles of intensive variables such as the single scattering albedo (SSA), lidar ratio (LR) or effective radius are assumed as vertically constant by this method. As a result, for each GRASPpac retrieval we obtain aerosol profiles (at 60 points) of backscatter at $180°$, scattering, extinction and absorption coefficients at 440, 675, 870, 1020 and 1064 nm, and also of aerosol size distribution (but without changes in the effective radius with height) and the aerosol volume concentration."

Comment: [5,21] "backscattering": here "under 180_"?

Answer: Yes, we have modified this.

Comment: [5,22] Please add a short comment on the accuracy of the retrieved aerosol parameters that are used in Section 4 (see also comment [3,1]). This is an important/mandatory information.

Answer: We have included a comment on the uncertainty of the retrieved parameters. Please, see our response to comment [3,1].

Comment: [5,26] "backscattering": here "hemispheric"?

Answer: Yes, we have changed it.

Comment: [5,30] Give an equation/definition of "comparison" and "relative difference" (see also [6,2] and figures): "GRASP minus in-situ" or "in-situ minus GRASP"? Divided by the "mean of in-situ"?

Answer: We have included the definition.

Comment: [6,7] Is the angular range with respect to the backscatter configuration correct? It should be 90° instead of 10° here (cf. Müller et al., 2011a)?

Answer: Yes, the design of the instrument limits the collection of the scattered light to the angular range 10-171°. However, we agree with the reviewer that this information here is misleading since after applying the correction introduced by Müller et al. (2011a) the scattering coefficient should correspond to the angular range 0-180°. With the aim of also clarifying how the comparison of the backscatter coefficient retrieved with GRASPpac and the hemispheric backscattering coefficient measured with the nephelometer this paragraph has been completely modified. Please, see our response to comment [4,2].

Comment: [6,12] "tends to overestimate all the studied...". Is this statement trivial?

Answer: We have modified this paragraph as follows:

"The frequency distributions of the absolute errors (in-situ minus GRASPpac values) for the scattering and extinction coefficients are tailed towards negative values evidencing

[Figure]

an overestimation of GRASPpac retrievals compared with in-situ measurements. For the extinction coefficient, Herreras et al. (2018) showed good agreement between the integrated extinction profiles derived with GRASPpac and AOD from sunphotometers located at various heights (R2>0.6). For the backscattering coefficient, Fig.2 shows that GRASPpac also overestimates the in-situ measurements, but the frequency distribution of the absolute errors is more symmetrically distributed around 0. The overestimation of GRASPpac retrieved backscattering coefficients is in agreement with the assumption made to convert the backscatter coefficient at 180° provided by GRASPpac into a hemispheric backscattering coefficient in order to perform the comparison with the in-situ measurements (see section 3). As the backscatter at 180° is typically larger than at smaller angles, this overestimation was expected. However, since overestimation of the total scattering and extinction coefficients also occurs, it is difficult to discern whether this overestimation originates in the GRASPpac retrieval or in the assumption made to compare with the in-situ data. On the other hand, this assumption might be contributing to lower the correlation between the backscattering coefficient from GRASPpac and in-situ measurements in comparison with the results obtained for the scattering and extinction coefficients comparison (Fig. 1), that shows higher correlation coefficients "

Comment: [6,22] "...there is a linear trend between scattering and extinction coefficients...": What does this mean? If scattering and extinction are the same, the single scattering albedo is = 1. This is more or less the case under clean and turbid conditions according to Fig. 3a. If scattering and extinction show a linear dependence, the single scattering albedo is constant. The fact that in general large extinction coefficients correspond to large scattering coefficients is not surprising (! is typically between say 0.8 and 1).

Answer: The idea behind this figure was not to show that there is a linear dependence between the scattering and extinction coefficients nor to show that large extinction coefficients correspond to large scattering coefficients, since as mentioned by the reviewer,

this is not surprising. With this figure, we wanted to show the different relationship between scattering and extinction for the in-situ measurements (single scattering albedo close to 1, perfect linear trend) compared to the GRASPpac retrievals. GRASPpac retrievals do not reproduce the observed behavior. The absorption coefficient is overestimated in many cases which leads to differences in the scattering-extinction pattern as it is observed from in-situ measurements. Due to the climatic relevance of the single scattering albedo, we think that the performance of GRASPpac retrieving absorption or single scattering albedo values is an important topic to discuss in the manuscript.

Comment: [8,17] "Qualitatively speaking, the volume concentration...". I don't understand the reason for this sentence? The seasonal cycle of the volume size distribution is not shown in the paper. Is this sentence included because a similar behavior is plausible? Or because this is known from literature? Or is it an intrinsic feature of the GRASP-methodology?

Answer: We included this sentence to note that the seasonality of the volume concentration and the scattering coefficient are similar to the one shown in the manuscript for the extinction coefficient. Although it was not shown in the paper, the volume concentration and scattering coefficient vertical profiles were also retrieved and available to perform the statistical analysis. To avoid confusion we have removed this sentence from the manuscript.

Comment: [8,19] A comment on the limitation to daytime: This is caused by the combination with the sun photometer. From the ceilometer's perspective the determination of backscatter coefficients can be provided during night time as well (likely even better) provided that a calibration is possible, see comment on [3,1]. The extinction coefficient can be estimated if the lidar ratio can be estimated (of course, subject to – maybe significant – errors). In Titos et al.'s paper the transformation from RCS to physical quantities is provided by using the photometer data as constraint; this is conceptually superior (at the expense of the two measuring systems required).

Answer: We agree with the reviewer. We have specifically noted that this limitation is caused by the combination with the sun/sky photometer instrument. See also our response to comment [3,1] on this topic. However, it is important to keep in mind that the use of the sun/sky photometer, although restricted to day-time, allows us to obtain profiles of more optical properties and not only backscatter, as well as microphysical aerosol properties.

Comment: [8,23] above -> larger than

Answer: Done

Comment: [10,1] Explain "center of mass"; e.g. by citing Mona et al. (2006) or alternatively (if you want to avoid another self-citation) Binietoglou et al. (2015).

Answer: The following sentence has been included:

"The center of mass gives in a single number an indication of the altitude of the aerosol vertical distribution in the atmosphere. Cases in which a single aerosol layer is present in the atmosphere, the center of mass gives an indication of its mean altitude; in cases of multiple layers, however, it could be located in areas without any considerable aerosol load (Binietoglou et al., 2015; Mona et al., 2006)."

Comment: [10,4] Here and in Fig. 7, the center of mass is given with respect to sea level. The numbers are correct but may lead to misunderstandings as the reader would intuitively expect much lower values (main contribution is almost always from the mixing layer). Indeed, values of 1.0–1.5 km can be found from Fig. 7 when considering heights above ground. So I recommend to add the corresponding values in brackets, at least in one or two cases.

Answer: We have included the height above the ground in this sentence and a specific statement in caption of Fig. 7 noting that heights refer to the sea level, but measurements start at 1570 m a.s.l.

Comment: [10,28] "Similar seasonal behavior...": This is not a result of this study, it is

only a message from a paper by Pandolfi et al. (2014). This should be made clear.

Answer: We agree with the reviewer, we have deleted this sentence from the conclusions.

Comment: [11,12] Here, Illingworth et al. (2018) can be cited.

Answer:Thank you for the reference, we have included it in the revised version of the manuscript.

Comment: [Fig. 5] delete "light" in the caption; also in Fig. 6.

Answer: Done

Comment: Suggested references:

Binietoglou, I., Basart, S., Alados-Arboledas, L., Amiridis, V., Argyrouli, A., Baars, H., Baldasano, J. M., Balis, D., Belegante, L., Bravo-Aranda, J. A., Burlizzi, P., Carrasco, V., Chaikovsky, A., Comerón, A., D'Amico, G., Filioglou, M., Granados-Munoz, M. J., Guerrero-Rascado, J. L., Ilic, L., Kokkalis, P., Maurizi, A., Mona, L., Monti, F., Munoz-Porcar, C., Nicolae, D., Papayannis, A., Pappalardo, G., Pejanovic, G., Pereira, S. N., Perrone, M. R., Pietruczuk, A., Posyniak, M., Rocadenbosch, F., Rodriguez-Gómez, A., Sicard, M., Siomos, N., Szkop, A., Terradellas, E., Tsekeri, A., Vukovic, A., Wandinger, U., and Wagner, J.: A methodology for investigating dust model performance using synergistic EARLINET/AERONET dust concentration retrievals, Atmos. Meas. Tech., 8, 3577-3600, https://doi.org/10.5194/amt-8-3577-2015, 2015.

Mona, L., A. Amodeo, M. Pandolfi, and G. Pappalardo (2006), Saharan dust intrusions in the Mediterranean area: Three years of Raman lidar measurements, J. Geophys. Res., 111, D16203, doi:10.1029/2005JD006569. Illingworth, A., Cimini, D., Haefele, A., Haeffelin, M., Hervo, M., Kotthaus, S., Löhnert, U., Martinet, P., Mattis, I., O'Connor, E., and Potthast, R.: How can Existing Ground- Based Profiling Instruments Improve European Weather Forecasts?, Bull. Amer. Meteorol. Soc., https://doi.org/10.1175/BAMS-D-17-0231.1, 2018.

Wiegner, M. and Geiß, A.: Aerosol profiling with the Jenoptik ceilometer CHM15kx, Atmos. Meas. Tech., 5, 1953-1964, https://doi.org/10.5194/amt-5-1953-2012, 2012.

Please also note the supplement to this comment:
https://www.atmos-meas-tech-discuss.net/amt-2018-431/amt-2018-431-AC3-supplement.pdf
* * *
[Figure]

**Supplement:**

[revised manuscript text omitted]

---

## Author Response (AR2)

We thank the Editor and the reviewers for their comments and suggestions that have helped to improve the quality of the manuscript. A point by point response to their comments is included below. The reviewer's comments are shown in bold. Changes in the manuscript are noted between quotation marks. The new version of the manuscript with the changes tracked is included in a separate file.

**Anonymous Referee #3**

**Two technical comments: the Illingworth-paper has been published meanwhile (update the citation).**

Done

**Page5, line 3 (of the "diff"-version): "greater than 85% at around 770 m a.g.l." maybe better: "greater than 85% beyond 770 m from the ceilometer" or something similar. An almost identical sentence (page 5, line 22) can be deleted or shortened."**

Done